# Analysis of Output Signal Distortion of Galvanic Isolation Circuits for Monitoring the Mains Voltage Waveform

**DOI:** 10.3390/s22207769

**Published:** 2022-10-13

**Authors:** Matouš Havrlík, Martin Libra, Vladislav Poulek, Pavel Kouřím

**Affiliations:** Department of Physics, Faculty of Engineering, Czech University of Life Sciences Prague, 165 00 Prague, Czech Republic

**Keywords:** line voltage, total harmonic distortion, THD, galvanic separation

## Abstract

Different methods for galvanically isolated monitoring of the mains voltage waveform were evaluated. The aim was to determine the level of distortion of the output signal relative to the input signal and the suitability of each method for calculating active power values. Six fixtures were tested: two voltage transformers, an electronic circuit with a current transformer, a standalone current transformer, a simple circuit with optocouplers, and a circuit with an A/D-D/A converter with capacitive coupling. The input and output waveforms were mathematically analyzed by three methods: (1) calculating the spectral components of waveforms and the relative changes in their THD (total harmonic distortion) values, (2) determining the similarity of waveforms according to the size of the area bounded by the input and output waveform curves, and (3) determining the accuracy of the active power calculation based on the output waveform. The time difference in the zero crossing of the input and output signals was measured, and further calculations for the second and third method were performed on the zero-crossing time shift-corrected waveforms. Other aspects of selecting the appropriate type of monitoring element, such as power consumption or overall circuit complexity, were also evaluated.

## 1. Introduction

The importance of renewable energy sources has been growing rapidly in recent years, and much effort has been devoted to increasing their efficiency [1]. Among alternative energy sources, small photovoltaic power plants on the roofs of houses operated by homeowners are becoming increasingly popular. In some cases, it makes sense for operators of these small PV systems to maximize their own solar energy consumption within the household rather than selling the unconsumed energy to the grid.

A special device (PV wattrouter [2]) can be used to control the energy consumption of a household with a PV source and prevent possible energy overflows into the distribution network. This device monitors the instantaneous values of the active power supplied to or drawn from the grid and controls the household energy consumption, for example, by switching resistive appliances appropriately. When designing a new type of wattrouter, we came across the need to accurately record voltage and current waveforms in the network. For the monitoring of the current waveform, a lead-through current transformer proved to be suitable. This method also ensured galvanic isolation of the mains from the low-voltage part of the current measuring circuit. To monitor the mains voltage waveform, it is necessary not only to reduce the mains voltage amplitude to a suitable value in the order of units of volts, but the low-voltage part of the circuit must also be galvanically isolated from the measured mains voltage for safety reasons. From a physical point of view, the galvanic isolation of the input signal corresponding to the mains voltage waveform can be implemented by several different methods with regard to the character of the transmitted signal. Since the monitored mains voltage waveform is AC, a transformer with galvanically isolated primary and secondary windings can be used. The signal transfer from input to output is based on the principle of electromagnetic induction. Another form of galvanically isolated signal transmission is optical transmission. The electrical input signal is converted by a suitable source into light, which after passing through an optical path is incident on a light detector, which converts the light signal into an electrical signal. An optocoupler with an LED [3] and a photodiode can be used to transmit the analogue signal [4]. Another optical method for measuring higher (AC) voltages uses the voltage-induced change in optical polarization in anisotropic crystals. A polarized beam of light passes through the crystal and the resulting polarization, determined by the magnitude of the voltage applied to the crystal, is detected by the sensor [5,6]. A capacitive coupling using a capacitor can be used for galvanically isolated information transfer. When properly connected, the capacitors block DC voltages and low frequency signals. Higher frequencies, which the capacitors passes, are used to transmit information. The waveform of the mains voltage with a first harmonic component at 50/60 Hz, together with possible higher harmonic components at multiple frequencies, is digitized, and the data transfer by capacitive coupling is carried out using a high frequency carrier signal. On the output side, the digital signal is converted back to an analogue waveform. A capacitive coaxial voltage probe (CVP) can also be used for non-contact voltage measurement in a conductor [7].

The analogue measuring instruments used until recently are being widely replaced by digital instruments. In some applications, simple data such as the effective value of the measured quantities are sufficient for others, more complex tasks are required. With the use of compact programmable microcontrollers, the application possibilities are greatly expanded. The input circuit of the above-mentioned device—the wattrouter—is an electronic wattmeter. The integrated microcontroller, after sampling several periods of mains voltage and current, calculates the instantaneous value of the active power supplied to/from the mains using an algorithm. The quality of the reproduction of the monitored waveforms can significantly affect the accuracy of the calculated power value. In terms of the accuracy requirements of the calculation, attention must be paid to the selection of suitable galvanically isolating elements/circuits, which can distort the input signal in various ways. Another factor that can adversely affect the quality of the signal waveform transmission is the increasing inharmonicity of the mains voltage. Since the mains voltage waveform with a fundamental harmonic component (50/60 Hz) is significantly distorted by a large number of higher harmonic components up to units of kHz [8,9], the frequency dependence of the transmission characteristics of some galvanically isolating elements/circuits can have a significant effect on the distortion of the output signal. In order to find a suitable element/circuit galvanically separating the measuring part of the wattmeter from the network, several representatives of different physical methods were selected. The most important parameter for comparison of the elements/circuits was the degree of distortion of the output signal with respect to the observed mains voltage waveform. However, attention was also paid to practical application aspects such as power consumption, circuit complexity and the level of unwanted components in the output signal.

## 2. Materials and Methods

Six different measurement fixtures were tested in terms of the distortion of the waveform of the output voltage signal with respect to the input voltage signal. Two small transformers with similar specifications and construction but diametrically different output signal distortion were tested: K-2803GCE-1FL and BV EI 305 2055 HAHN. The ZMPT101B single-phase AC voltage sensor consisting of a 2 mA:2 mA current transformer and an electronic circuit was the next product tested. The ZMPT101B current transformer was also tested separately, without electronic circuit, as another fixture. To test the method of optical galvanically isolated voltage signal transmission, pairs of LEDs and photodiodes in HCPL-4562 integrated circuits were used. The last product tested was the fixture with AMC1211 isolated amplifier circuit based on the principle of an A/D-D/A converter with capacitive digital signal transfer. The galvanically separating elements and fixtures are shown in Figure 1.

The first tested fixture was a 230 V transformer K-2803GCE-1FL designed for low-voltage power supplies, shell type with winding on a two-chamber frame with a EI metal sheet core with dimensions 30 × 14 × 27 mm (length, width, height) and a transformation ratio of 1:14.7. Mains voltage was applied directly to the primary winding terminals. The unconnected transformer is shown in Figure 1a. The input and output signals were monitored at the outputs of the resistive dividers connected to the primary and secondary windings (see Figure 2).

The second tested fixture was a 230 V transformer BV EI 305 2055 HAHN designed for low-voltage power supplies, shell type with winding on a two-chamber frame with a EI metal sheet core with dimensions 32.8 × 27.8 × 29.5 mm (length, width, height) and transformation ratio 1:21.6. Mains voltage was applied directly to the primary winding terminals. The output signal was measured on one of the two secondary windings and the other remained unloaded. The unconnected transformer is shown in Figure 1b. The input and output signals were monitored at the outputs of the resistive dividers connected to the primary and secondary windings (see Figure 3).

The third fixture tested was the ZMPT101B single-phase AC voltage sensor. It is a printed circuit board (PCB) on which a precision transformer of the same designation is mounted, guaranteeing galvanic isolation. On the primary side of the transformer, there is a biasing resistor (R13) limiting the current flowing through the primary side of the transformer. On the secondary side, there is a measuring resistor (R1). The analogue output signal from the ZMPT101B transformer enters the operational amplifier circuits where the signal is filtered, amplified and passed to the output. The ZMPT101B module allows adjustment of the output signal gain using an integrated multi-turn potentiometer. The mains voltage applied to the input terminals of the fixture was monitored at the output of the resistive divider and the unipolar output signal was recorded at the output pin (OUT) of the fixture. The device needs to be powered on the low-voltage side using a stabilized 5 V supply. The maximum mean voltage value at the primary side is 250 VAC/50 Hz. A photo of the module is shown in Figure 1c. The wiring of the module and its internal diagram is shown in Figure 4.

The fourth tested fixture was a stand-alone current transformer ZMPT101B. The transformer consists of galvanically isolated primary and secondary windings with a 1:1 ratio wound on a toroidal ferrite core. The line voltage was applied to the primary winding through a 150 kΩ limiting resistor. The mains voltage applied to the input terminals of the fixture was monitored at the output of the resistive divider, and the bipolar output signal was recorded on the measuring resistor 500 Ω. The unconnected standalone current transformer is shown in Figure 1d. The wiring diagram of the fixture is shown in Figure 5.

The fifth fixture tested was a simple circuit with two optocouplers consisting of LED–photodiode pair. HCPL-4562 integrated circuits with LED and photodiode connected to the transistor base were selected. However, the current output of the photodiode was used directly for the measurement. In order to process both positive and negative half wave of the input AC 230 V signal, the LEDs were connected antiparallel with common current limiting resistors. The antiparallel connection of the LEDs also ensured the limitation of the shutter voltage on the LEDs. The size of the input limiting resistors was chosen by the mean current of the LEDs corresponding to their nominal average forward current value of 12 mA. Each of the photodiodes was connected to its own reverse voltage source, and the bipolar output signal was read at the measuring resistor 20 kΩ. An operational amplifier connected as a voltage follower was inserted between the measuring resistor and the fixture output. The fixture with the optocouplers connected is shown in Figure 1e. Although the circuit was implemented on a solderless breadboard, the nature of the output signal was not affected, but the use of a printed circuit board and soldering of components may result in less signal noise. The wiring of the fixture can be seen in Figure 6.

The sixth fixture tested was the AMC1211 isolated amplifier circuit from Texas Instruments and is based on the principle of an A/D and D/A converter that is integrated directly into the AMC1211 circuit. The analogue low-voltage signal from the input resistive divider is processed by the integrated A/D converter. This processed digital signal is transferred through an isolation barrier to the low-voltage part of the circuit where the D/A converter converts the digital signal back to an analogue signal. This signal is then amplified using an LM358 operational amplifier and sent to the output terminal of the fixture. The integrated circuit AMC1211 allows processing the signal in the range of 0–2 V. The value of the resistive divider is calculated in such way that the output of the divider is a maximum of 2 V even in case of overvoltage in the grid. The resistive divider is designed using precision MELF type resistors with a percentage error of 0.1–1%. The AMC1211 integrated circuit requires separate power supplies for the low- and high-voltage parts. The power supply was provided by 9 V batteries and LM7805 linear regulators. The shift in the analogue output signal from the resistive divider to positive values for the possibility of processing the entire waveform by the AMC1211 integrated circuit is provided on the fixture by the AP7115 linear regulator, which shifts the signal by exactly 1 V. This shift ensures that the signal entering the AMC1211 integrated circuit does not contain a negative component and can be processed completely. A fixture with an AMC1211 circuit to monitor the mains voltage and a current transformer to measure the current to/from the mains designed for the “PV wattrouter” device is shown in Figure 1f. The circuit diagram of the fixture with the resistive divider and the AMC1211 integrated circuit is shown in Figure 7.

The diagram of the measuring apparatus is shown in Figure 8. The mains voltage was galvanically isolated from the measuring apparatus by a transformer with separate primary and secondary windings with a transformation ratio of 1:1. A Thalheimer LTS 604 adjustable laboratory transformer was used. Since it included a regulating autotransformer, the output mean voltage value was set to 230 V. The input and output voltage waveforms were simultaneously recorded using a 16-bit A/D converter Data Translation DT9816 with six single earth analog input channels. The measuring range of the converter was −10 V to +10 V. Input AC signals with peak values of ±325 V were reduced approximately to ±3.5 V by a resistive divider. The parameters of the output voltage signal varied from fixture to fixture. The output waveform of both transformers tested did not contain a DC component, the peak voltage values were ±22.1 V and ±15.0 V, respectively, and a resistive divider was used to record the output signal so that the values did not exceed ±5 V. Peak output signal values from other fixtures did not exceed ±5 V.

For the mathematical processing, sample records of ten whole signal periods of the mains voltage waveform and the output voltage waveform from the fixture were used. Source signals of 50 Hz frequency were sampled with the frequency of 750,000 samples per second. Three different methods were used to evaluate the effect of the fixture on the distortion of the output signal relative to the input waveform.

### 2.1. Comparison of the Relative Change in Total Harmonic Distortion of the Voltage Waveforms (THDu)

The first evaluation method consisted of comparing the *THDu* (Total Harmonic Distortion of voltage waveform) [10] of the input and output signals. First, a spectral frequency analysis of both signals was performed with a 1 Hz step. To find the amplitude of each frequency component, the Fourier transformation [11] applied to the whole signal recording (ten whole periods), mathematically modulated by a Hanning window [12] of width corresponding to the whole recording period (ten periods), was used. The assumption that the dominant components of the frequency spectrum were integer multiples of the fundamental frequency of 50 Hz was confirmed. For the frequencies corresponding to the harmonic components of the signals, a more accurate calculation of the amplitudes and phase angles of all components was performed using the original signal, undistorted by application of the Hanning window. The amplitude values of the first 40 harmonics of the frequency spectrum were used to calculate the *THDu* using Equation (1) [13]
(1)$$THDu=\frac{\sqrt{V_{2}^{2}+V_{3}^{2}+V_{4}^{2}+\ldots +V_{n}^{2}}}{V_{1}}\cdot 100\;\left[\%\right]$$
where *V_x_* represents the value of the amplitude of the *x*-th harmonic component of the voltage signal. Since the mains voltage signal itself is already significantly distorted due to non-linear loads connected to the network [14,15,16], the increment of *THDu* of the output signal compared to *THDu* of the input signal was determined as
(2)$$\Delta THDu=THDu_{out}-THDu_{in}$$

The distortion of the harmonic waveform of the network voltage was not constant and the *THDu* of the input signal varied from measurement to measurement, the relative increase in the *THDu* with respect to the *THDu* of the input was chosen as the evaluation criterion
(3)$$relative\;increase\;in\;THDu=\frac{\Delta THDu}{THDu_{in}}\cdot 100\;\left[\%\right]$$

The phase shift in the output signal relative to the input voltage waveform was determined as the phase shift in the first harmonic components based on frequency spectral analysis. The phase shift in the instant of signal zero crossings was determined manually. If the output signal precedes the input signal, the phase shift values are negative, if the output signal is delayed after the input signal, the values are positive. However, the objectivity of this method of evaluating the quality of individual fixtures is limited by the fact that the mains voltage does not have a purely harmonic waveform, is distorted by higher harmonics, and some fixtures exhibit frequency dependence of the transfer characteristic [17,18].

### 2.2. Comparison of Output Voltage Waveform Distortion Using the Ratio of the Area Bounded by the Input and Output Waveforms Relative to the Area Bounded by the Input Waveform and the Time Axis

The second evaluation method compares the degree of distortion in the output signal of each fixture by comparing the shape differences in the input and output waveforms. The DC components were mathematically removed from the waveforms in case they contained this component. Then, the values of the curves were normalized according to their maximum value. For noisy or interfered signals, the waveform was normalized by the maximum value of the floating-averaged data. The pairs of curves being compared were mathematically phase shifted so that the zero-crossing moments were the same for both curves. The *difference area* between the two curves can be calculated as
(4)$$difference\;area=\frac{\sum_{s=1}^{n}\left|u_{in}\left(s\right)-u_{out}\left(s\right)\right|}{n}\cdot t$$

The area bounded by the input signal curve and the *t*-axis can be expressed as
(5)$$input\;signal\;area=\frac{\sum_{s=1}^{n}\left|u_{in}\left(s\right)\right|}{n}\cdot t$$

The value of *t* corresponds to the recording duration (*t* = 10∙*T* = 0.2 s), *s* is the sample order and *n* is the total number of samples in the record (*n* = 150,000).

The percentage value of the shape distortion of the output waveform “*difference area error*” was obtained as the ratio of the difference area value to the input signal area:(6)$$difference\;area\;error=\frac{difference\;area}{input\;signal\;area}\cdot 100\left[\%\right]=\frac{\sum_{s=1}^{n}|u_{in}\left(s\right)-u_{out}\left(s\right)|}{\sum_{s=1}^{n}\left|u_{in}\left(s\right)\right|}\cdot 100\left[\%\right]$$

If the output signal was significantly affected by noise or interference, a mathematically adjusted data signal was used to extend the comparison of distortion for the second method. The data were processed by taking a moving average of nine values according to the formula:(7)$$\bar{u}\left(t\right)=\frac{\sum_{s=t-4}^{t+4}u\left(s\right)}{9}$$

### 2.3. Comparison of Output Signal Waveform Distortion in Terms of Active Power Calculation Accuracy

The third evaluation method of comparing the quality of the fixture output was chosen for the case where the only evaluation criterion would be the accuracy of the calculation of the active power. The DC components were mathematically removed from the waveforms if they contain this component and then the values of the curves were normalized according to their maximum value. For noisy or interfered signals, the waveform was normalized by the maximum value of the floating-averaged data. The input and output waveforms were mathematically phase shifted so that the zero-crossing moments were the same for both curves. The active power component is proportional to the mean value of the square of the voltage values for the entire recording time (*t* = 10∙*T* = 0.2 s):(8)$$P_{in,out}=constant\cdot \frac{\sum_{s=1}^{n}u_{in,out}^{2}\left(s\right)}{n}$$

The error in the active power calculation based on the output signal from the fixture relative to the actual active power is obtained as
(9)$$active\;power\;error=\left(1-\frac{P_{out}}{P_{in}}\right)\cdot 100\%$$

## 3. Results and Discussion

The input and output signal record of the K-2803GCE-1FL transformer is shown in Figure 9a). A comparison of the input and output signal waveforms normalized by their maximum values with a corrected zero-crossing offset is shown in Figure 9b).

The input and output signal record of the BV EI 305 2055 transformer is shown in Figure 10a). A comparison of the input and output signal waveforms normalized by their maximum values with a corrected zero-crossing offset is shown in Figure 10b).

The input and output signal record of the ZMPT101B single-phase AC voltage sensor is shown in Figure 11a). A comparison of the input and output signal waveforms normalized by their maximum values with a corrected zero-crossing offset is shown in Figure 11b).

The input and output signal record of the ZMPT101B stand-alone current transformer is shown in Figure 12a). A comparison of the input and output signal waveforms normalized by their maximum values with a corrected zero-crossing offset is shown in Figure 12b).

The input and output signal record of the fixture with optocouplers HCPL-4562 is shown in Figure 13a). A comparison of the input and output signal waveforms normalized by their maximum values is shown in Figure 13b). Type-matched integrated circuits were used to monitor the mains voltage during the positive and negative half-periods. However, the electrical parameters of the integrated circuits slightly differed in the signal gain. The corresponding coefficient values for the positive and negative half-periods were used to normalize the output signal. No zero-crossing time delay was observed between the input and output signals. 

The transfer characteristics of the input and output current of the optocouplers for both input signal polarities were measured (see Figure 14). It can be seen that in neither case is the dependence linear. The use of an LED–photodiode pair proved inappropriate for this reason. A slightly more complex circuit with two photodiodes optically coupled to a single input LED and circuitry limiting nonlinearity [19,20] would have been a more appropriate choice.

The threshold voltage of the LEDs used was approximately 1.1 V. Due to the maximum mains voltage of around 325 V, the unmeasured area of the voltage waveform should represent only 0.3% of the entire range of values. The effect of the delayed LED opening is harder to observe on the output signal graph because there are short-term oscillations of the output signal in the near-zero voltage region, which are probably caused by the operational amplifier. However, from the plot of the difference in the instantaneous values of the input and output signals (see below, Section 3.2), the jump in the waveform in the time region around the zero crossing is clearly visible. The magnitude of this jump corresponds to approximately ±1.5% of the mains voltage amplitude. These values could be used to correct the output waveform values to obtain a more accurate result.

The disadvantage of this circuit was the high power consumption (2.75 W) of the fixture due to thermal losses on the resistors limiting the input LED current. Reducing the input power led to a reduction in the signal-to-noise ratio of the output signal.

The input and output signal record of the fixture with AMC1211 is shown in Figure 15a). A comparison of the input and output signal waveforms normalized by their maximum values is shown in Figure 15b). No zero-crossing time delay was observed between the input and output signals.

When measuring the fixture with AM1211 in the laboratory, interference always appeared in the output signal. With several repetitions, the nature of the interference changed. The signal-to-interference ratio could be increased by simply increasing the gain of the output amplifier.

### 3.1. Comparison of the Relative Change in Total Harmonic Distortion of the Voltage Waveforms (THDu)

The amplitudes of the first 40 harmonic components of both the input and output voltage signals were calculated to compare the distortion of the voltage waveform by each fixture. Table 1 shows the amplitudes of the most significant components of the frequency spectrum—the first five odd harmonic frequencies—normalized by the corresponding value of the amplitude of the first harmonic component.

For each fixture, the *THDu* values of the recorded mains voltage waveforms and the output signal from the fixture were calculated. The relative increase in the *THDu* of the output to the *THDu* of the input signal was used for comparison. Furthermore, the phase shift in the input and output signals was determined in terms of the phase shift in the first harmonic component. Values are shown in Table 2. The phase shift in the input and output signals determined in terms of the signal zero crossing is shown in Table 3.

The AMC1211 isolated amplifier circuit has the lowest relative change in *THDu*. The ZMPT101B stand-alone transformer reduced the *THDu*. In the case of ZMPT101B with electronic circuit, the result sharply contrasted with the results found by the second and third methods. Apparently it was heavily distorted shape of waveform compared to the input signal and it did not show the expected increase in *THDu*. The cause was probably the frequency dependence of the fixture’s transfer characteristic, but this was not investigated in this paper. As an objective method, comparison of the relative change in THDu could only be used if the input signal contained only one harmonic component, namely, the fundamental harmonic component at 50/60 Hz.

### 3.2. Comparison of Output Voltage Waveform Distortion Using the Ratio of the Area Bounded by the Input and Output Waveforms Relative to the Area Bounded by the Input Waveform and the Time Axis

The differences in instantaneous values of normalized and time-corrected (the same instant of the signal passing through zero) input and output signals are obtained by:(10)$$\;value\;difference=u_{in}\left(s\right)-u_{out}\left(s\right)$$
which are used for subsequent calculation, and are shown for one period (20 ms) of records for transformer K-2803GCE-1FL, ZMPT101B single-phase AC voltage sensor, and optocouplers (see Figure 16).

A comparison of all test fixtures in terms of the ratio of the area bounded by the input and output waveforms relative to the area bounded by the input waveform and the time axis can be seen in Table 4.

The ZMPT101B current transformer followed by the fixture with AMC1211 show even the smallest deviations determined by this method. The output signals from the ZMPT current transformer and optocouplers were noisy, and the output signal for the AMC1211 isolated amplifier circuit was strongly affected by high-frequency sweeps repeating at a frequency of approximately 25 kHz. For these three fixtures, the signal was further adjusted with a moving average, and the difference area error calculation was repeated. With this modified signal, the AMC1211 isolated amplifier circuit has the lowest distortion in terms of difference area error.

### 3.3. Comparison of Output Signal Waveform Distortion in Terms of Active Power Calculation Accuracy

In order to compare the fixtures in terms of accuracy of the calculation of active power, the ratio of the power corresponding to the output signal waveform from the fixture to the actual power corresponding to the real mains voltage waveform was calculated for all fixtures. The values determined by Equation (9) and reported in Table 5 were then compared.

The smallest absolute values of error in the calculation of active power using the fixture output data were achieved with the AMC1211 isolated amplifier circuit. The transformer BV EI 305 2055 achieved only a slightly worse result. The small error of the result for the voltage transformer BV EI 305 2055 is the result of compensating the overestimation of the instantaneous power values in the first and third quarter of the signal period and the underestimation of the power in the second and fourth quarter of the period.

Table 6 provides a summary of the calculated errors of the individual fixtures by all three methods and phase shift in the zero crossing of the output signal relative to the input signal. The power consumption of the input (high-voltage) parts of the circuits was determined and included in the table.

Both voltage transformers have an input consumption determined by their design. The input consumption of the assembled AC voltage sensor with ZMPT is determined by the size of the limiting resistor in series with the primary winding of the current transformer. For a fixture with a separate ZMPT current transformer, the value of the limiting resistor was chosen so that the effective input current was approximately halfway through its recommended range of 1–2 mA. The value of the power dissipation at the maximum allowed input current is then shown in brackets. In the case of the fixture with optocouplers, the limiting resistors set the nominal photodiode current, which is not allowed to be exceeded. The value of the power dissipation is therefore the same as the maximum value shown in brackets. For the fixture with an Amc1211 integrated circuit, the input power is determined by the choice of the total resistance of the input divider. This circuit needs also its own low-voltage power supply circuit on the input side, whose consumption will vary from type to type and is therefore not taken into account.

## 4. Conclusions

Six different fixtures for mains voltage waveform monitoring with galvanically isolated output were compared. The following fixtures were tested: voltage transformers K-2803GCE-1FL and BV EI 305 2055, the ZMPT101B single-phase AC voltage sensor, stand-alone current transformer ZMPT101B, optocouplers HCPL-4562 and the AMC1211 isolated amplifier circuit.

The selection of the appropriate element/circuit will be determined by the requirements of the application.

In terms of accurate reproduction of the mains voltage waveform, less than 2% error was achieved by fixtures with a ZMPT stand-alone current transformer, a pair of optocouplers and an AMC1211 AD/DA converter.

In terms of the accuracy of the active power calculation, fixtures with a ZMPT stand-alone current transformer and a pair of optocouplers achieved errors of less than 2% and fixtures with a BV EI 305 2055 voltage transformer and an AMC1211 AD/DA converter achieved errors of less than 1%.

The K-2803GCE-1FL voltage transformer proved to be unsuitable in terms of output signal distortion, unlike voltage transformer BV EI 305 2055, as the second representative of the electromagnetic galvanic isolation principle of similar design and basic parameters. In principle, the voltage transformer is suitable for monitoring the mains voltage, but care must be taken in selecting its specific type.

Especially in small applications, the energy consumption of the type of separating element/circuit can play an important role. Most suitable elements tested had a maximum power consumption on the mains side of approximately 1 W, the exception of the tested fixtures being the circuit with optocouplers. However, this circuit can be implemented with input-side power consumption less than 1 W, but signal reproduction accuracies for smaller input currents were not determined in this paper. The fixture with the AMC1211 AD/DA converter had the lowest power consumption on the mains side, but it needs a separate power supply for the high-voltage part of the circuit.

The fixture with an AMC1211 AD/DA converter performed well in terms of signal distortion evaluation by all methods used, but it is a more complex circuit. It turned out that the tested circuit was also not very robust to interference, its output was noisier, and the circuit would need to be further modified. The second, more complex circuit tested was an AC voltage sensor with a ZMPT 101B current transformer. This circuit is industrially manufactured and could provide a cheap complete solution for monitoring mains voltage, but it did not achieve sufficient accuracy in reproducing the input signal, probably due to high THDu. The other tested fixtures consisted basically only of a galvanic isolation module and voltage dividers; the operational amplifier connected as a voltage follower was used only for the needs of reading the output signal waveform using an AD converter in a circuit with optocouplers. The voltage and current transformers did not need any additional power supplies to obtain the output bipolar signal. The single-channel bipolar output of the optocoupler pair circuit required two independent low-voltage DC sources. The circuit with optocouplers can be implemented as a two-channel with one source common to both channels.

For some fixtures, the zero-crossing time difference between the input and output signals was detected. This parameter, however, was not a determining factor in the evaluation of signal distortion, as it was assumed that its value was in-variant and could be used for mathematical phase correction of the output waveform during signal processing. The need for this computational correction could be limiting for some applications, and methods with little phase shift (ZMPT stand-alone current transformer) or no phase shift (optocouplers, AMC1211 AD/DA converter) would be more suitable.

The first method of evaluating the output signal distortion was the comparison of the relative change in total harmonic distortion of the voltage waveforms (*THDu*). It turned out that the result can be strongly influenced by the fact that the input signal is not purely harmonic, but is more or less distorted by a large number of harmonic components. The *THDu* of the network, caused mainly by connected non-linear loads, fluctuates over time, and the frequency spectrum of the monitored voltage can change strongly. The frequency dependence of the tested elements/circuits and the effect of the *THDu* change on the accuracy of the required measurement outputs should also be determined, but this was not investigated in this work.

## Figures and Tables

**Figure 1 sensors-22-07769-f001:**
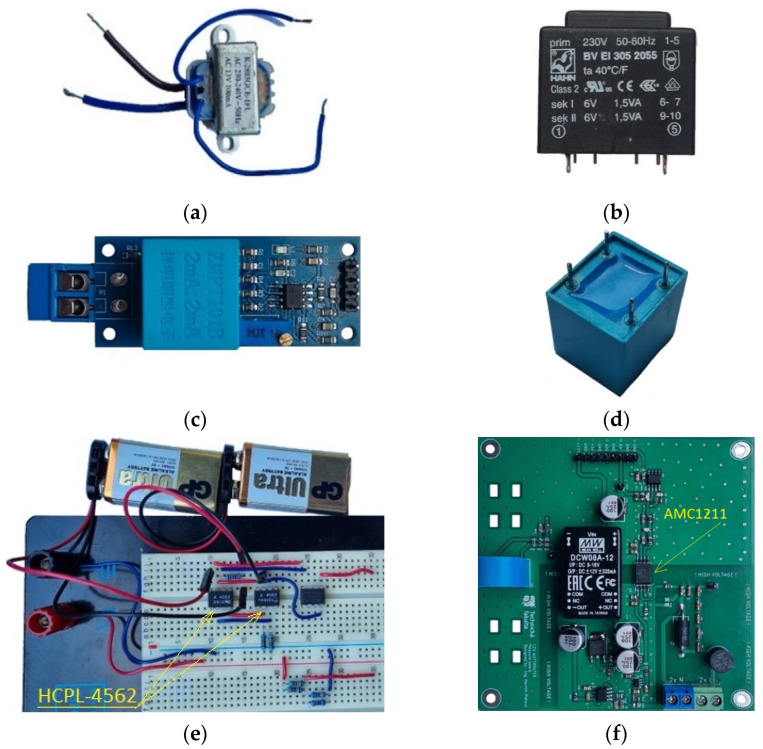
(**a**) K-2803GCE-1FL transformer, (**b**) BV EI 305 2055 HAHN transformer, (**c**) ZMPT101B single-phase AC voltage sensor, (**d**) Stand-alone current transformer ZMPT101B, (**e**) The fixture with the optocouplers HCPL-4562, (**f**) The fixture with an AMC1211 circuit.

**Figure 2 sensors-22-07769-f002:**
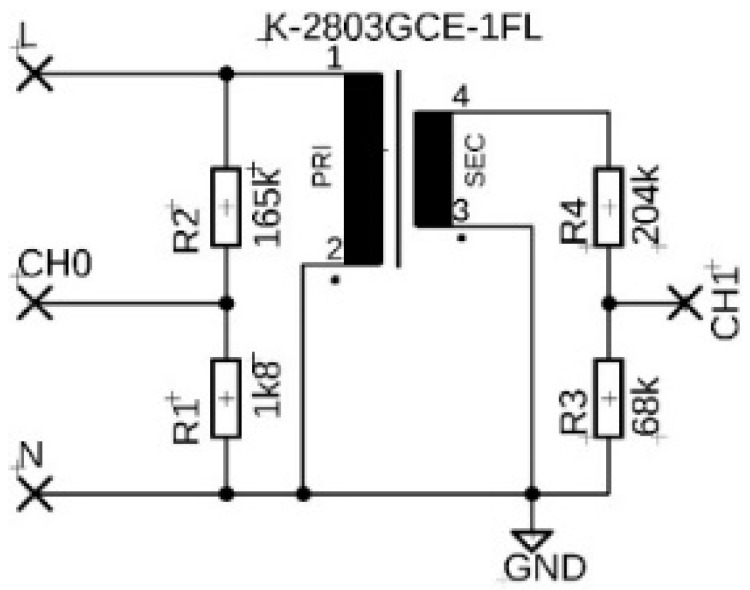
Measuring circuit diagram with K-2803GCE-1FL transformer.

**Figure 3 sensors-22-07769-f003:**
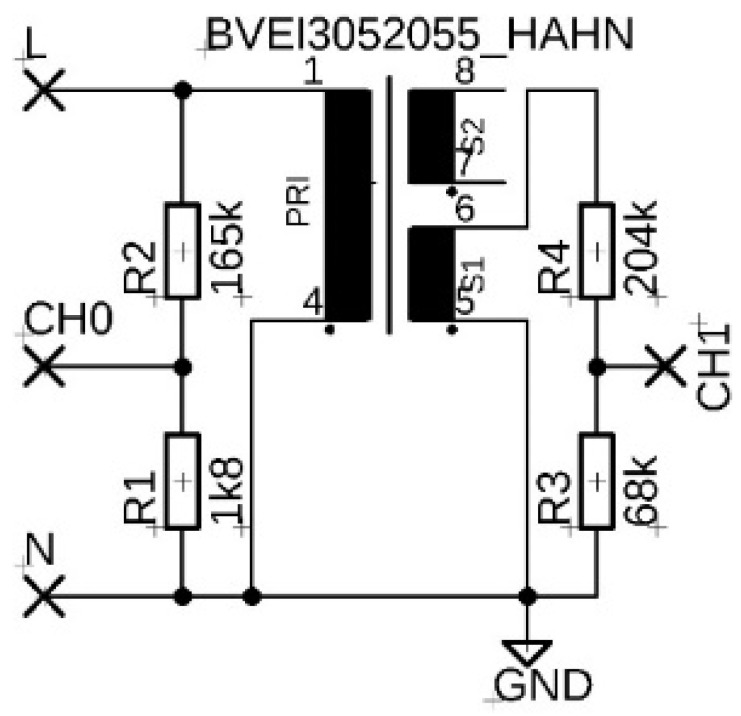
Measuring circuit diagram with BV EI 305 2055 HAHN transformer.

**Figure 4 sensors-22-07769-f004:**
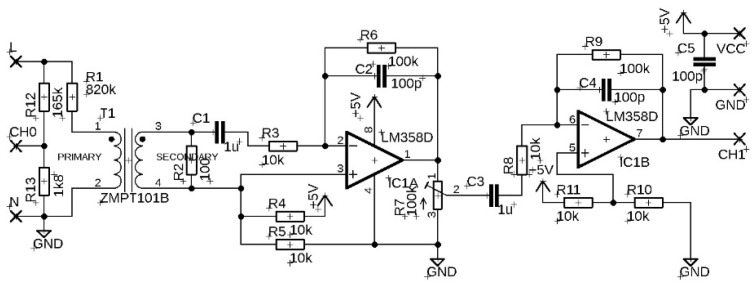
The wiring of the ZMPT101B single-phase AC voltage sensor and its internal diagram.

**Figure 5 sensors-22-07769-f005:**
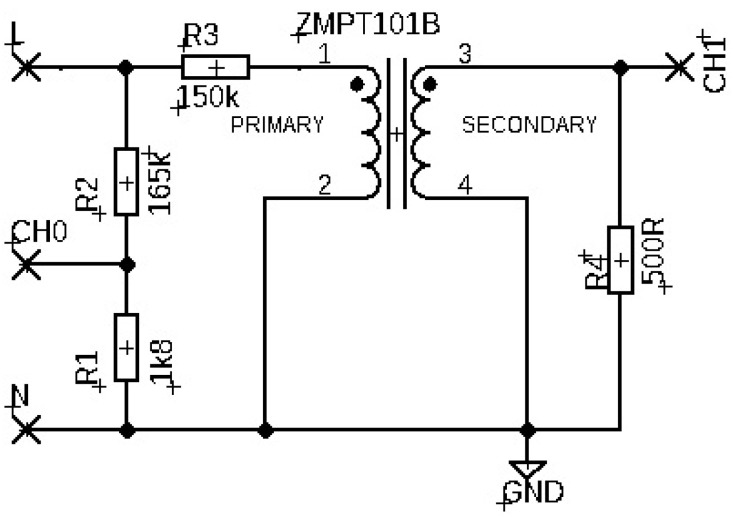
The wiring diagram of the fixture with the stand-alone current transformer ZMPT101B.

**Figure 6 sensors-22-07769-f006:**
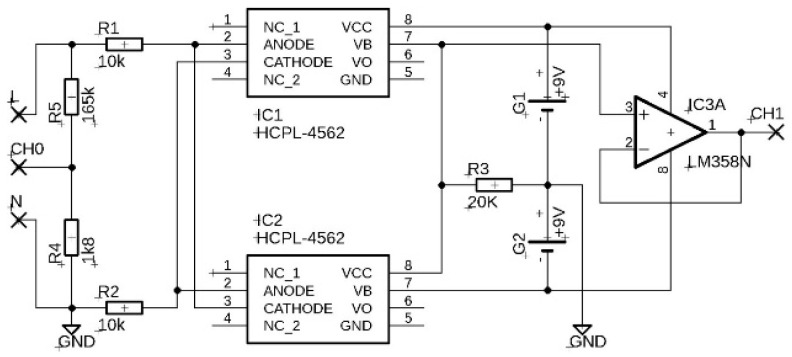
The wiring of the fixture with the optocouplers HCPL-4562. HCPL-4562 circuit pins: 1-not connected, 2-LED anode, 3-LED cathode, 4-not connected, 5-transistor emitter, 6-transistor collector, 7-transistor base and photodiode anode, 8-photodiode cathode.

**Figure 7 sensors-22-07769-f007:**
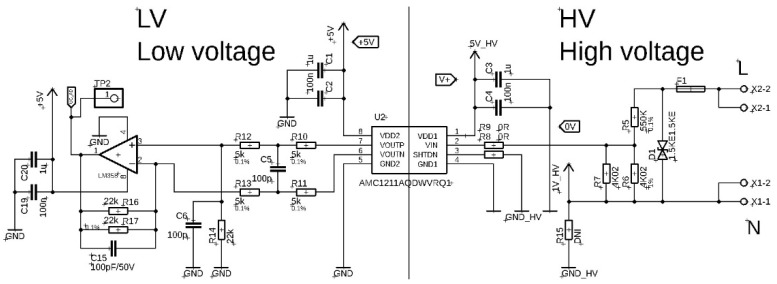
The circuit diagram of the fixture with the resistive divider and the AMC1211 integrated circuit.

**Figure 8 sensors-22-07769-f008:**
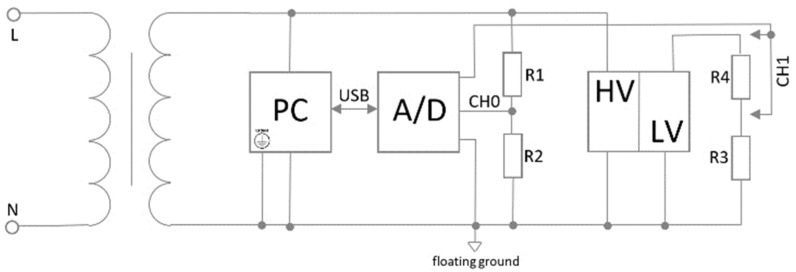
The wiring diagram of the measuring apparatus. The Live (L) and Neutral (N) contacts of mains are connected to the primary winding of the isolation transformer. One of the outputs of the isolation transformer secondary winding is selected as a common floating ground for grounding the PC, A/D converter, high-voltage (HV) and low-voltage (LV) sides of the fixtures and resistive dividers. If the maximum output voltage from the fixture was higher than the range of the AD converter, the signal for CH1 was sensed at the voltage divider (R3, R4).

**Figure 9 sensors-22-07769-f009:**
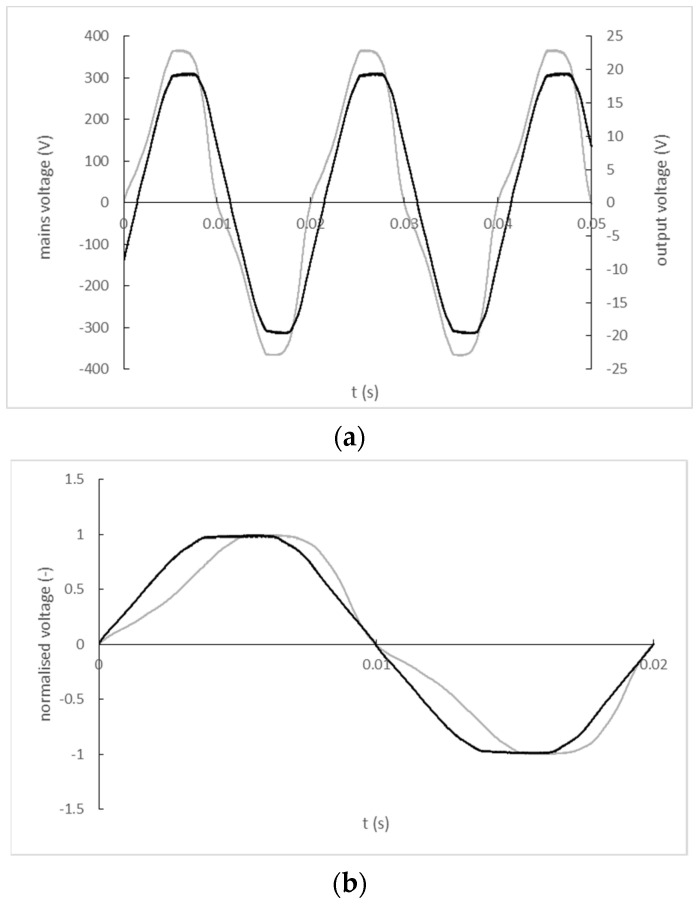
(**a**) The input and output signal record of the K-2803GCE-1FL (black line—mains; grey line—output); (**b**) Normalized and zero-crossing time shift-corrected input and output waveforms of K-2803GCE-1FL transformer (black line—mains; grey line—output).

**Figure 10 sensors-22-07769-f010:**
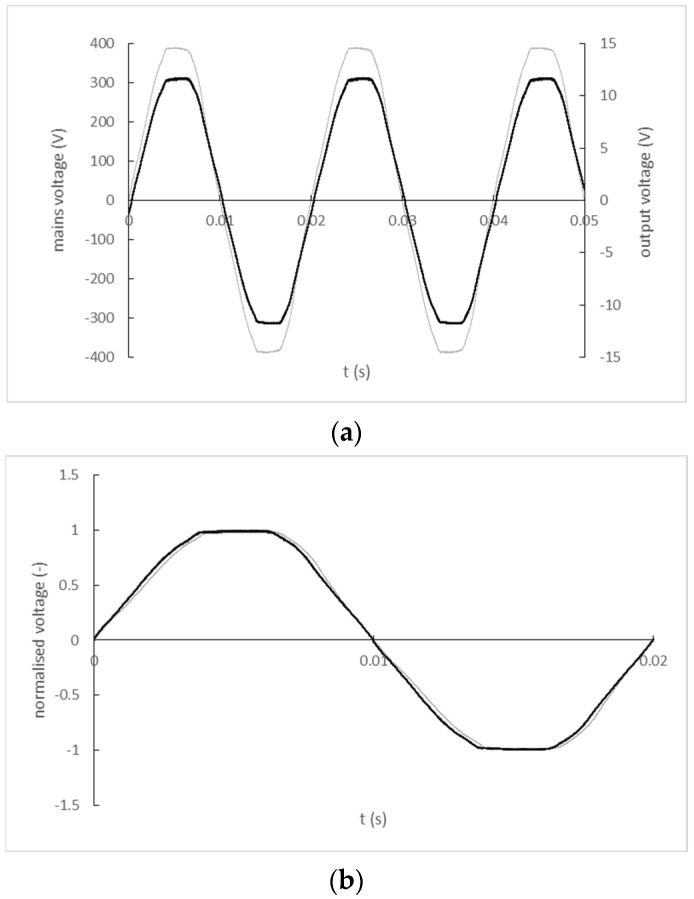
(**a**) The input and output signal record of the BV EI 305 2055 transformer (black line–mains, grey line–output); (**b**) Normalized and zero-crossing time shift-corrected input and output waveforms BV EI 305 2055 transformer (black line—mains; grey line—output).

**Figure 11 sensors-22-07769-f011:**
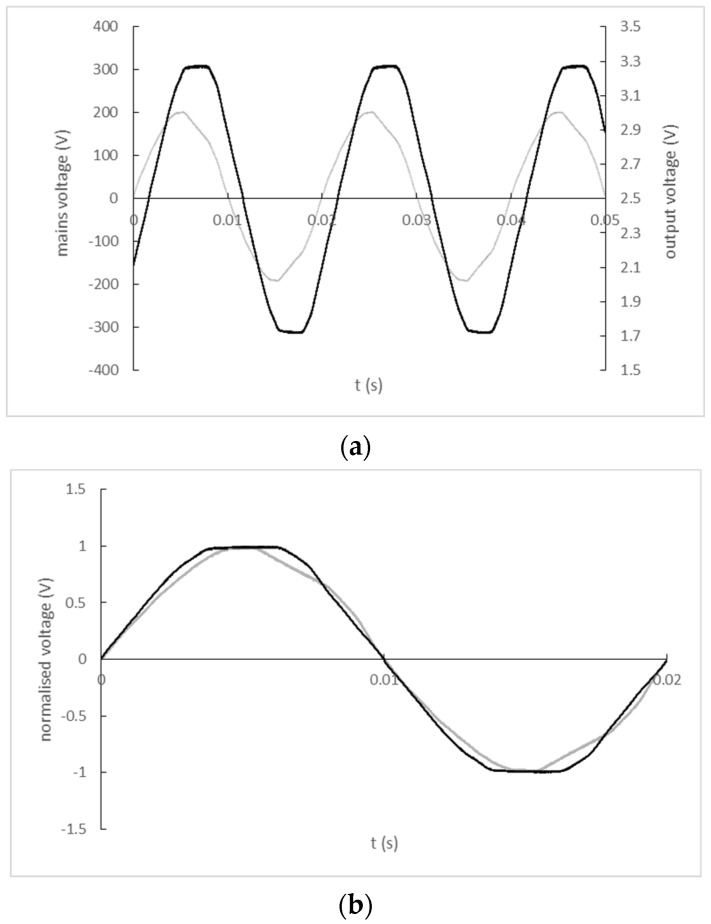
(**a**) The input and output signal record of ZMPT101B single-phase AC voltage sensor (black line—mains; grey line—output); (**b**) Normalized and zero-crossing time shift-corrected input and output waveforms of ZMPT101B single-phase AC voltage sensor (black line—mains; grey line—output).

**Figure 12 sensors-22-07769-f012:**
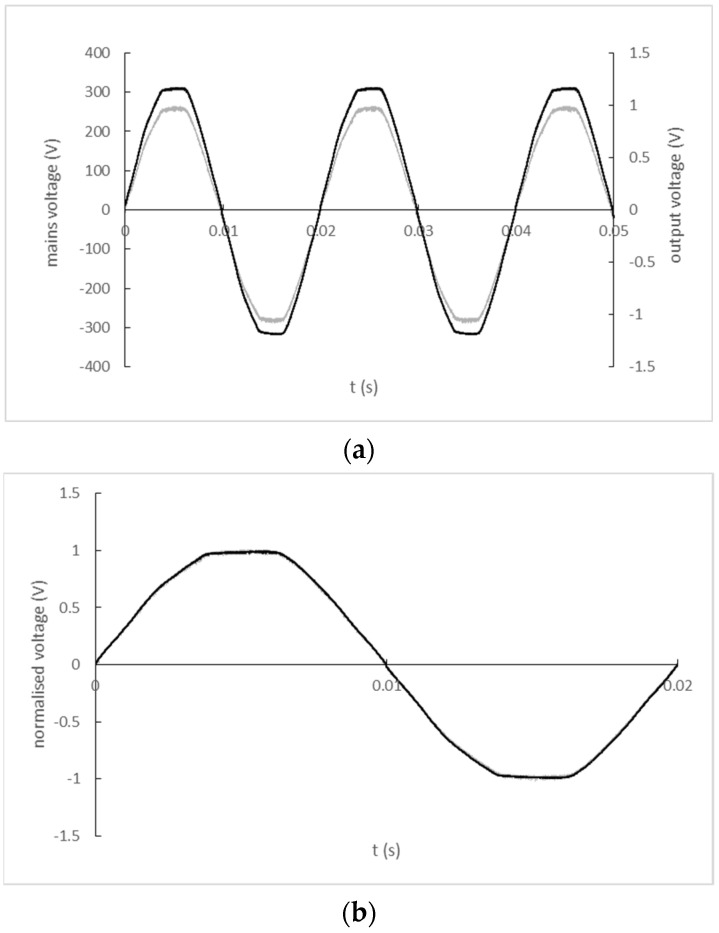
(**a**) The input and output signal record of ZMPT101B current transformer (black line—mains; grey line—output); (**b**) Normalized and zero-crossing time shift-corrected input and output waveforms of ZMPT current transformer (black line—mains; grey line—output).

**Figure 13 sensors-22-07769-f013:**
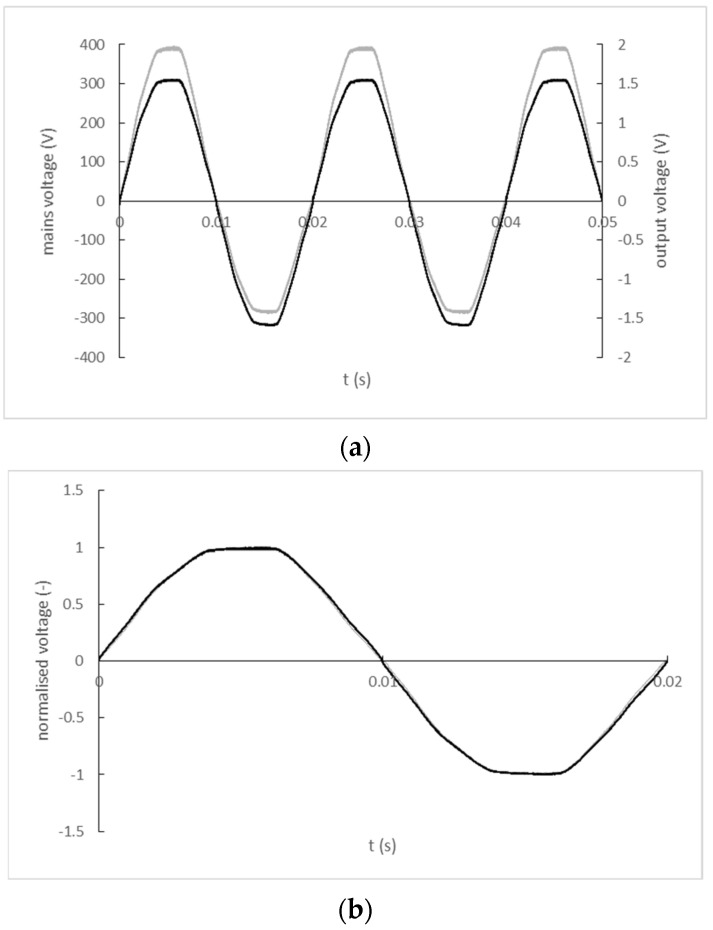
(**a**) The input and output signal record of optocouplers HCPL-4562 (black line—mains; grey line—output); (**b**) Normalized input and output waveforms of optocouplers HCPL-4562 (black line—mains; grey line—output).

**Figure 14 sensors-22-07769-f014:**
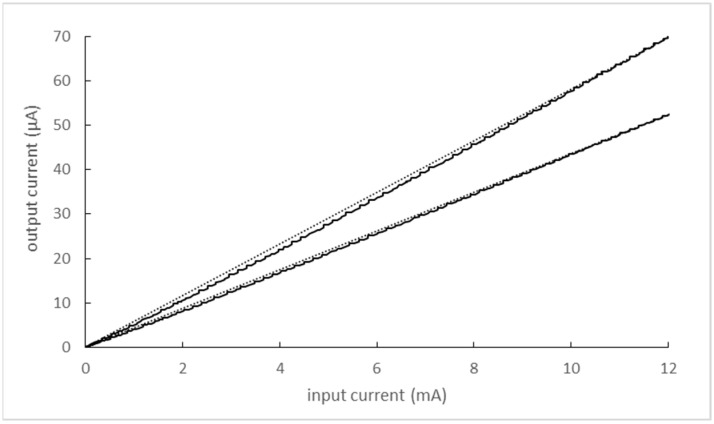
The transfer characteristics of the input and output currents of two optocouplers HCPL-4562 (black lines—characteristics of both optocouplers; dotted lines—linear dependencies of input and output quantities given by corresponding end points of optocoupler characteristics).

**Figure 15 sensors-22-07769-f015:**
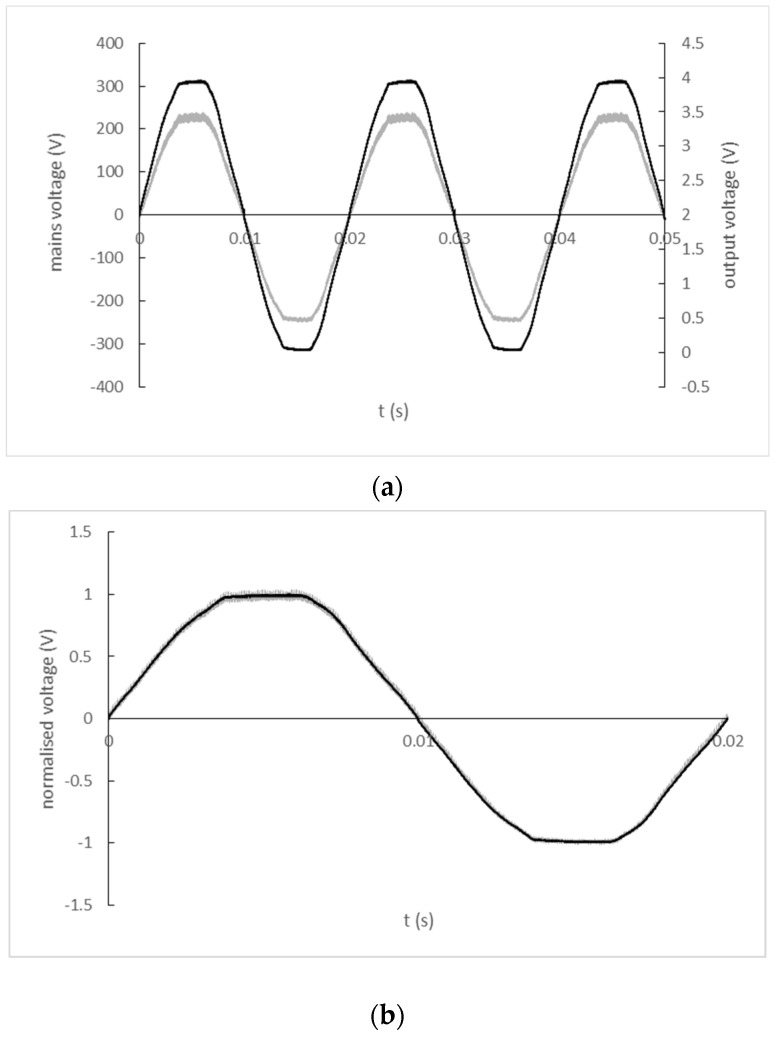
(**a**) The input and output signal record of AMC1211 isolated amplifier (black line—mains; grey line—output); (**b**) Normalized input and output waveforms of AMC1211 isolated amplifier (black line—mains; grey line—output).

**Figure 16 sensors-22-07769-f016:**
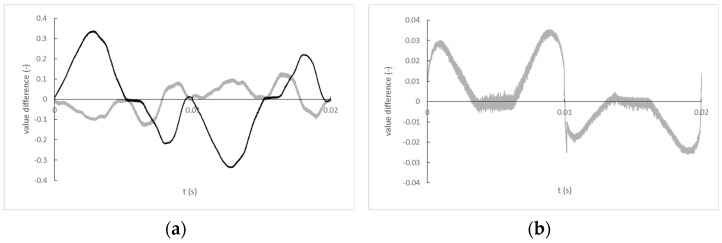
(**a**) Differences in instantaneous values of normalized and time-corrected input and output signals of one period for transformer K-2803GCE-1FL (black line) and ZMPT101B single-phase AC voltage sensor (grey line); (**b**) Differences in instantaneous values of normalized and time-corrected input and output signals of one period for fixture with HCPL-4562 optocouplers. The sharp jump in the plotted values near the zero crossing of the input signal (0, 0.01, 0.02 s) is clearly visible.

**Table 1 sensors-22-07769-t001:** Amplitudes of the first five odd harmonic components of the frequency spectrum of the waveforms normalized by the value of the amplitude of the first harmonic component.

		K-2803GCE-1FL		BV EI 305 2055		ZMPT101B on PCB		ZMPT101B		HCPL-4562		AMC1211	
Order	f(Hz)	Mains	Output	Mains	Output	Mains	Output	Mains	Output	Mains	Output	Mains	Output
1	50	1.000	1.000	1.000	1.000	1.000	1.000	1.000	1.000	1.000	1.000	1.000	1.000
3	150	0.014	0.134	0.013	0.029	0.013	0.013	0.012	0.011	0.011	0.011	0.014	0.014
5	250	0.030	0.062	0.028	0.034	0.028	0.031	0.019	0.018	0.020	0.028	0.027	0.028
7	350	0.008	0.024	0.009	0.007	0.010	0.010	0.008	0.007	0.010	0.007	0.011	0.011
9	450	0.001	0.006	0.002	0.002	0.001	0.001	0.007	0.007	0.007	0.010	0.002	0.002

**Table 2 sensors-22-07769-t002:** *THDu* of mains and output signal, increment of *THDu*, relative increase in *THDu*, and a phase shift in the first harmonic component of input and output waveforms.

	K-2803GCE-1FL	BV EI 305 2055	ZMPT101B on PCB	ZMPT101B	HCPL-4562	AMC1211
THDu of mains [%]	3.4	3.3	3.3	2.6	2.7	3.3
THDu of fixture output [%]	14.9	4.6	3.6	2.4	3.3	3.5
increment of THDu [%]	11.5	1.3	0.3	−0.2	0.7	0.1
relative increase in THDu [%]	336.9	39.4	8.3	−7.8	25.5	4.5
phase shift in 1st harm. comp. [°]	−14.4	−3.8	−28.1	−0.4	−0.2	0.0

**Table 3 sensors-22-07769-t003:** The phase shift in the signal zero crossing of the input and output signals.

	K-2803GCE-1FL	BV EI 305 2055	ZMPT101B on PCB	ZMPT101B	HCPL-4562	AMC1211
phase shift in zero crossing [°]	−25.8	−5.9	−29.5	−0.4	0.0	0.0

**Table 4 sensors-22-07769-t004:** The difference area error values calculated according to Equation (6) for the normalized input and output waveforms with time-corrected source data are in the row labelled “difference area error”. The difference area error values for output waveforms for which noise or interference has been reduced by averaging according to Equation (7) are shown in the row labelled “with noise reduction”. (NC—not calculated).

	K-2803GCE-1FL	BV EI 305 2055	ZMPT101B on PCB	ZMPT101B	HCPL-4562	AMC1211
difference area error [%]	21.534	4.061	8.331	1.329	1.885	1.659
with noise reduction [%]	NC	NC	NC	1.033	1.829	0.746

**Table 5 sensors-22-07769-t005:** The active power error values calculated according to Equation (9) for the normalized input and output waveforms with time-corrected data.

	K-2803GCE-1FL	BV EI 305 2055	ZMPT101B on PCB	ZMPT101B	HCPL-4562	AMC1211
active power error [%]	11.200	0.544	11.288	1.461	1.598	−0.209

**Table 6 sensors-22-07769-t006:** Summary of the calculated errors and the energy consumption of the input parts of the circuits. Difference area error values in parentheses correspond to the output signal corrected by the floating average calculation according to Formula (7). The input power loss values in parentheses correspond to the maximum adjustable input current.

	K-2803GCE-1FL	BV EI 305 2055	ZMPT101B on PCB	ZMPT101B	HCPL-4562	AMC1211
relative increase in THDu [%]	336.9	39.4	8.3	−7.8	25.5	4.5
difference area error, (NR) [%]	21.534	4.061	8.331	1.329, (1.033)	1.885, (1.829)	1.659, (0.746)
active power error [%]	11.2	0.544	11.288	1.461	1.598	−0.209
phase shift in zero crossing [°]	−25.8	−5.9	−29.5	−0.4	0	0
input-side power loss, (max) [W]	2.56	1.07	0.07	0.35, (0.46)	2.76, (2.76)	0.1

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
