# Peer review of "Analysis of Output Signal Distortion of Galvanic Isolation Circuits for Monitoring the Mains Voltage Waveform"

_sensors, 2022, doi:10.3390/s22207769_

Round 1
Reviewer 1 Report
The comparison methodology presented in this paper is not a good idea for the assembled electronic circuit or module, or single element. This is because of those in the following.
a) The electronic elements, like Figs. 1(a) and (b), are generally different in practical applications. Only their output signals taken for comparison is not fair.
b) The testing device shown in Fig. 1(f) is obviously a commercial module (with commercial PCB). This usually gives a better performance in all signals measured from the components. Fortunately, this has been confirmed in the testing result in this study.
c) Why was the solderless breadboard used for the circuit building? as shown in Fig. 1(e).
From the above review issues, it is difficult for readers to know what conclusions are given from this study.
Author Response
The authors are very grateful to all the reviewers for their time spent on the revision and for their valuable advice that improved the quality of the article. We have revised the article and the changes can be viewed in the "all revisions" mode. Below is a detailed response to each comment.
Reviewer 1:
The comparison methodology presented in this paper is not a good idea for the assembled electronic circuit or module, or single element. This is because of those in the following.
- a) The electronic elements, like Figs. 1(a) and (b), are generally different in practical applications. Only their output signals taken for comparison is not fair.
When testing several transformers, large differences in the level of output signal distortion were observed. For this reason, we included two voltage transformers of similar parameters in the study, but one with large and the other with very small distortion. Both transformers are primarily designed for low-voltage power supplies of small power (units of watts). Both are shell type with windings on a double chamber frame, the core is made of EI shaped plates. The difference in output signal distortion is probably due to the difference in core material. The relevance and reason for the comparison of these transformers is explained by the added comments in Introduction and Conculusions.
- b) The testing device shown in Fig. 1(f) is obviously a commercial module (with commercial PCB). This usually gives a better performance in all signals measured from the components. Fortunately, this has been confirmed in the testing result in this study.
In the construction of the new type of "wattrouter", a module for monitoring the mains voltage and current was designed and manufactured with an integrated circuit AMC1211, which promised good output signal parameters. This prototype was tested as the sixth fixture (shown in Figure f) in the study. Since this module is quite complex and requires an extra separate power supply on the 230V side and hence more expensive, other options suitable for possible use in a commercial product were sought. The other methods tested for monitoring the mains voltage and circuit with AMC1211 became the basis of this study.
- c) Why was the solderless breadboard used for the circuit building? as shown in Fig. 1(e).
Yes, it's true that the optocoupler was to be implemented as a PCB-based circuit for this study. The new fixture with optocouplers on a printed circuit board was made with optional wiring (with or without operational amplifier, with 20kΩ or 64kΩ input limiting resistors, with or without shifting the bipolar output signal to positive values).
Detailed results were not processed due to time constraints. The waveforms for the original size of the limiting resistors (20kΩ) at the input but without the op amp are very similar to the original. Waveforms of the circuit with smaller input power (0.83W, 64kΩ limiting resistors) without operational amplifier connected as a voltage follower cannot be read correctly by AD converter, although the recording on oscilloscope proves usability of this circuit as well. An operational amplifier is required here…
From the above review issues, it is difficult for readers to know what conclusions are given from this study.
All selected representatives of different methods of galvanic separation and stress reduction except fixture 3) can be considered as basic elements. The circuit with optocouplers contains an operational amplifier, but it is connected only as a voltage follower to eliminate the jump in output values from the AD converter when the input signal passes through zero. The AMC1211 integrated circuit cannot work without support circuits, but the circuit diagram is very simple here as well. The only exception is the ZMPT101B single-phase AC voltage sensor, which contains both a galvanic isolation element and an add-on circuit. It was included among the elements tested as it appeared to be a possible suitable low-cost complex solution. The stand-alone ZMPT current transformer was tested separately.

Reviewer 2 Report
The paper stuides analysis of output signal distortion of galvanic isolation circuits for monitoring the mains voltage waveform.
To improve this paper, I have some comments as follows:
(1) For introduction section, Literature review should be more detailed and comprehensive. Authors should add more recent research progress to this part and give a brief introduction of the development history on your topic.
(2) It is better for authors to clarify your objectives of your study in the Introduction section.
(3) For your proposed method, advantages compared to other methods should be clarified.
(4) Reasons why you choose this method should be added to your manuscript.
(5) In the comparative study, authors can use other methods to analyze the same problem and highlight your method's accuracy.
(6) For conclusions, more detailed results should be presented.
Author Response
The authors are very grateful to all the reviewers for their time spent on the revision and for their valuable advice that improved the quality of the article. We have revised the article and the changes can be viewed in the "all revisions" mode. Below is a detailed response to each comment.
Reviewer 2:
The paper stuides analysis of output signal distortion of galvanic isolation circuits for monitoring the mains voltage waveform.
To improve this paper, I have some comments as follows:
(1) For introduction section, Literature review should be more detailed and comprehensive. Authors should add more recent research progress to this part and give a brief introduction of the development history on your topic.
Several references have been added. The Introduction chapter has been extended.
(2) It is better for authors to clarify your objectives of your study in the Introduction section.
The aim of the study was described in more detail in the Introduction chapter.
(3) For your proposed method, advantages compared to other methods should be clarified.
(4) Reasons why you choose this method should be added to your manuscript.
(3) + (4) Although the circuit with the AMC1211 performs well in all evaluation methods used, the aim of the study was not to show this type of monitoring method as the most suitable for all applications. Other evaluation aspects relevant to monitoring method selection with respect to intended use have been included in the text.
(5) In the comparative study, authors can use other methods to analyze the same problem and highlight your method's accuracy.
To compare the distortion of the output signal shape, the first choice was the standard method used in the field of energy engineering to test the distortion of the waveforms, namely the calculation of the THD of the analyzed signals. Unfortunately, it turned out that the very existence of voltage waveform distortion in the network affects the resulting signal distortion given the nature of the fixtures. For fixtures with a frequency-dependent transfer characteristic, this evaluation method is not objective, but because it points to an important phenomenon, it is included in the study. We did not find any methods other than those used in the study suitable for comparing the outputs of preparations.
(6) For conclusions, more detailed results should be presented.
In the conclusion chapter, the individual fixtures are discussed in more detail.

Reviewer 3 Report
The proposed evaluation of the different methods for galvanically isolated monitoring of the mains voltage waveform is very interesting for the hardware engineers and scientists.
This paper should be added with the list of the measurement equipment and detailed description of the causes of the errors.
Also, it will be better to compare obtained results with the theoretical basis and show comparison results of the different circuits with the all parameters in one table.
Figure 7 should be simplified.
Author Response
The authors are very grateful to all the reviewers for their time spent on the revision and for their valuable advice that improved the quality of the article. We have revised the article and the changes can be viewed in the "all revisions" mode. Below is a detailed response to each comment.
Reviewer 3:
The proposed evaluation of the different methods for galvanically isolated monitoring of the mains voltage waveform is very interesting for the hardware engineers and scientists.
This paper should be added with the list of the measurement equipment and detailed description of the causes of the errors.
Only the AD converter mentioned in the text was used for the actual measurements. Its description has been clarified. Other necessary equipment such as oscilloscope multimeters and RLC meter were not used in the actual experiment.
The physical causes of errors in individual preparations were not specified due to time constraints.
Also, it will be better to compare obtained results with the theoretical basis and show comparison results of the different circuits with the all parameters in one table.
A study of the theory explaining the causes of the phenomena causing the distortion in the various methods used was not undertaken due to time constraints.
A summary table with the results of all three methods was added.
Figure 7 should be simplified.
Figure 7 has been simplified (supporting circuit diagrams have been removed).

Round 2
Reviewer 1 Report
The testing by measuring the signals from the breadboard-built circuit and the commercial PCB-based module is quite unfair. Maybe this can be a special reference for electronic engineers.
Author Response
The authors are very grateful to all the reviewers for their time spent on the revision and for their valuable advice that improved the quality of the article. We have revised the article and the changes are visible in mode „All revisions“. Below is a detailed response to each comment.
Reviewer 1:
The testing by measuring the signals from the breadboard-built circuit and the commercial PCB-based module is quite unfair. Maybe this can be a special reference for electronic engineers.
The use of solderless breadboard for optocouplers has been commented in the Materials and Methods chapter. The described nature of the distortion of the output signal from the circuit on the solderless breadboard was confirmed by a basic comparison of the waveforms obtained from the new fixture with the circuit on the printed circuit board.
Reviewer 2:
The newly added references 5 and 7 should be replaced as there are a lot of newer works in this field.
Reference 5 has been replaced by a more recent article describing the method. One more recent reference has been added [7]. The reviewer did not specify a specific article that should be cited. So we searched for and found an article in Sensors journal as well. Original reference [7] has been replaced by the designation of the latest version of the European Standard BS EN 50160:2010+A3:2019 [8].

Reviewer 2 Report
The newly added references 5 and 7 should be replaced as there are a lot of newer works in this field.
Author Response

(The authors gave the same response as above.)
